# Nitric Oxide Production from Nitrite plus Ascorbate during Ischemia upon Hippocampal Glutamate NMDA Receptor Stimulation

Carla Nunes [1,2,*] and João Laranjinha [1,2]

[1] Center for Neuroscience and Cell Biology, University of Coimbra, 3004-504 Coimbra, Portugal
[2] Faculty of Pharmacy, University of Coimbra, 3000-548 Coimbra, Portugal
* Correspondence: cmnunes@cnc.uc.pt

**Abstract:** Nitric oxide ($^\bullet$NO), a diffusible free radical, is an intercellular messenger, playing a crucial role in several key brain physiological processes, including in neurovascular coupling (NVC). In the brain, glutamatergic activation of the neuronal nitric oxide synthase (nNOS) enzyme constitutes its main synthesis pathway. However, when oxygen ($O_2$) supply is compromised, such as in stroke, ischemia, and aging, such $^\bullet$NO production pathway may be seriously impaired. In this context, evidence suggests that, as already observed in the gastric compartment, the reduction of nitrite by dietary compounds (such as ascorbate and polyphenols) or by specific enzymes may occur in the brain, constituting an important rescuing or complementary mechanism of $^\bullet$NO production. Here, using microsensors selective for $^\bullet$NO, we show that nitrite enhanced the $^\bullet$NO production in a concentration-dependent manner and in the presence of ascorbate evoked by N-methyl-D-aspartate (NMDA) and glutamate stimulation of rat hippocampal slices. Additionally, nitrite potentiated the $^\bullet$NO production induced by oxygen-glucose deprivation (OGD). Overall, these observations support the notion of a redox interaction of ascorbate with nitrite yielding $^\bullet$NO upon neuronal glutamatergic activation and given the critical role of NO as the direct mediator of neurovascular coupling may represents a key physiological mechanism by which $^\bullet$NO production for cerebral blood flow (CBF) responses to neuronal activation is sustained under hypoxic/acidic conditions in the brain.

**Keywords:** nitric oxide; nitrite; ascorbate; hippocampal slices





## 1. Introduction

Although the human brain represents only 2% of body weight, it consumes approximately 20% of the oxygen ($O_2$) and 25% of glucose utilized by the body and receives nearly 15% of the cardiac output [1]. These high energetic and metabolic demands of the human brain, associated with its very limited reserve capacity, imply a continuous and tightly regulated cerebral blood flow (CBF) to assure a proper supply of metabolic substrates, namely $O_2$ and glucose, as well as the clearance of metabolic waste byproducts [2]. In this context, it is important to note that even a small reduction in CBF has a deleterious impact on brain function [3,4]. As such, a mechanism termed neurovascular coupling (NVC), which involves a tight network communication between all the cells that comprise the Neurovascular Unit (neurons, glia and cerebrovascular cells), ensures a fine temporal and regional regulation of the CBF according to neuronal activity to fulfill the metabolic needs [5]. Recently, strong evidence has emerged that the highly diffusible free radical, nitric oxide ($^\bullet$NO), plays a pivotal role in the NVC [6–8]. In the Central Nervous System (CNS), $^\bullet$NO is also involved in other brain physiological processes, such as learning and memory formation, synaptic plasticity, mitochondrial respiration and modulation of neurotransmitter release [9–13].

The canonical pathway for $^\bullet$NO synthesis in mammalian cells is carried out by a family of enzymes, the nitric oxide synthases (NOS) [14]. These enzymes catalyze the

oxidation of L-arginine to L-citrulline and $^{\bullet}$NO, using NADPH and oxygen ($O_2$) as co-substrates [14]. There are three major isoforms of NOS: neuronal NOS (nNOS or NOS 1), endothelial NOS (eNOS or NOS 3) and inducible NOS (iNOS or NOS 2), all of them present in the CNS. Both nNOS and eNOS are constitutively expressed and activated by $Ca^{+2}$/calmodulin-dependent signaling, producing nanomolar concentrations of $^{\bullet}$NO for seconds or minutes [15]. iNOS is induced in glial cells following immunological or inflammatory stimulation, and its activity is $Ca^{2+}$-independent, producing high concentrations of $^{\bullet}$NO for hours or days [15].

$^{\bullet}$NO signaling in the brain is intimately associated with glutamatergic neurotransmission [16]. In neurons, the activation of N-methyl-D-aspartate receptor (NMDAr) by glutamate binding leads to the influx of $Ca^{+2}$ that, upon binding to calmodulin, activates nNOS, which is physically coupled to NMDAr by the scaffolding protein postsynaptic density-95 (PSD-95) [17]. In addition, at the capillary level, glutamate may activate the NMDAr in the endothelial cells and thus leads to eNOS activation.

As NOS requires $O_2$ to work properly, in situations where a decrease in $O_2$ supply occurs, such as stroke, ischemia, and aging, the activity of constitutive NOS is compromised, and thereby, the enzymatic $^{\bullet}$NO production can significantly decrease, leading to an impairment of NVC and other central $^{\bullet}$NO functions. However, in recent years, some studies point to nitrite as a key bioprecursor of $^{\bullet}$NO in the brain, particularly under acidic and hypoxic conditions [18–21], as it was already verified in the gastric compartment [22,23]. In this context, it was demonstrated that acute nitrite enhanced basal CBF in a rat model [24] and also recovered NVC to its original magnitude in a rat model of somatosensory stimulation under conditions of NOS inhibition [25].

Notably, nitrite concentration in the body may be enhanced through the diet, namely by ingesting green leafy vegetables, which are rich in nitrate [26]. Nitrate is quickly absorbed across the upper gastrointestinal tract [27,28]. Although much of the nitrate may be excreted in the urine, up to 25% is taken up by salivary glands, where it is concentrated and released into the oral cavity. Here, nitrate is reduced to nitrite by oral commensal bacteria [29]. In the stomach, a nitrite may be reduced to $^{\bullet}$NO [30,31]. Components of the diet, such as ascorbate and polyphenols, may enhance nitrite reduction [22,23]. The remaining nitrite is absorbed in the small intestine, and through circulation, it can reach several organs [18], including the brain, where increased nitrite levels are observed in the cerebrospinal fluid [32]. A growing body of evidence supports that the reduction of nitrite to $^{\bullet}$NO may be a possibility in the brain, namely by powerful reducing agents such as ascorbate and polyphenols [20,33], especially under low oxygen tension. In accordance, Presley and colleagues showed that a high-nitrate diet increases regional brain perfusion in older subjects in specific brain areas [34].

In this work, using microsensors with a high degree of selectivity to $^{\bullet}$NO, we explore the non-NOS production of $^{\bullet}$NO via nitrite reduction in rat hippocampal slices stimulated with NMDA and glutamate and also subjected to oxygen-glucose deprivation (OGD).

## 2. Materials and Methods

### 2.1. Materials and Reagents

NMDA was purchased from Hello Bio (Bristol, UK). Ortho-phenylenediamine (O-PD), Nafion®, ascorbic acid, L-glutamic acid and sodium nitrite were from Sigma-Aldrich. A Carbox gas mixture ($O_2$/$CO_2$) was obtained from Linde, Lisbon, Portugal. All solutions were prepared in MilliQ water, bi-deionized water ultrapure with resistivity higher than 18 MΩ.cm (Millipore Corporation, Burlington, MA, USA).

For hippocampal slice assays, we used artificial cerebrospinal fluid (aCSF) with the following composition (in mM): 124 NaCl, 2 KCl, 25 NaHCO3, 1.25 $NaH_2PO_4$, 1.5 $CaCl_2$, 0.1 $MgCl_2$ and 10 D-glucose. In order to improve the viability of the slices, a modified aCSF was used for slice dissection and recovery. The composition of this aCSF was (in mM): 124 NaCl, 2 KCl, 25 NaHCO3, 1.25 $NaH_2PO_4$, 1.5 $CaCl_2$, 1 $MgCl_2$, 1 reduced glutathione

(GSH), 0.2 ascorbate and 10 D-glucose. Both aCSFs were continuously bubbled with Carbox for oxygenation and pH buffering (pH 7.4).

## 2.2. Electrochemical Instrumentation

All recordings were performed using a Compactstat Potentiostat (Ivium, Eindhoven, The Netherlands). For $^\bullet$NO and $O_2$ hippocampal recordings, a two-electrode circuit was used with the microsensor as the working electrode and an Ag/AgCl (3M NaCl) as a reference electrode. The working electrode was held at a constant potential of +0.7 V or $-0.8$ V vs. Ag/AgCl for $^\bullet$NO and $O_2$ measurements, respectively.

## 2.3. Carbon Fiber Microelectrode Fabrication and Surface Modification

Carbon fiber microelectrodes (CFM) were essentially fabricated as previously described [35]. Briefly, a single carbon fiber ($\varnothing = 30$ µm, Textron, Lowell, MA, USA) was inserted into a borosilicate glass capillary (Science Products GmbH, Hofheim, Germany). The capillaries were pulled on a vertical puller (Harvard Apparatus Ltd., Cambourne, UK), and then the protruding carbon fibers were cut under a microscope to obtain a tip length of 150–200 µm. The electrical contact was provided by injecting a small portion of a conductive silver paint (RS Pro, Corby, UK) into the capillary, followed by the insertion of a copper wire with the outer insulation previously removed from the extremities. The microelectrodes were tested for their general recording properties in phosphate-buffered saline (PBS) medium by fast cyclic voltammetry (FCV) at a 200 V/s scan rate between –1.0 and 1.0 V vs. Ag/AgCl for 30 s (EI400 potentiostat, Ensman Instruments, Bloomigton, USA). The microelectrodes that passed in FCV evaluation (stable background current and sharp transients at reversal potentials) were modified in a two-step protocol with Nafion$^\circledR$ and O-PD to improve their analytical properties for $^\bullet$NO measurements. First, the tips of the CFMs were immersed in Nafion$^\circledR$ solution (5% in aliphatic alcohols) for ten seconds, followed by drying at 180 °C for 5 min. High temperatures seem to enhance the adherence of Nafion$^\circledR$ film to the carbon fiber surface [36]. After cooling, the process was repeated. On the day of the use, the CFMs were coated with O-PD by electropolymerization at a constant potential of +0.7 V vs. Ag/AgCl for 30 min.

## 2.4. Carbon Fiber Microelectrodes Calibration

Each $^\bullet$NO microelectrode was evaluated for $^\bullet$NO sensitivity and selectivity towards nitrite and ascorbate by constant voltage amperometry at +0.7 V vs. Ag/AgCl.

The CFMs for $O_2$ measurement were evaluated regarding their sensitivity towards $O_2$. After deoxygenation of PBS (40 mL) by bubbling nitrogen in a sealed vessel, several additions of 200 µL of an $O_2$-saturated solution, which corresponds to 6.22 µM of $O_2$, were performed. The sensitivity of the CFMs for $O_2$ was determined based on the slope of the calibration curve.

## 2.5. Rat Hippocampal Slices

All animal procedures used in this study were performed in accordance with the European Union Council Directive for the Care and Use of Laboratory Animals, 2010/63/EU, and were approved by the local ethics committee (ORBEA) and the Portuguese Directorate-General for Food and Veterinary. Male Wistar rats with ages between 6–7 weeks were decapitated under deep anesthesia. The brain was quickly removed and placed in ice-cold modified aCSF saturated with Carbox. The hippocampi were dissected, and transverse slices with a thickness of 400 µm were obtained with a McIlwain tissue chopper (Campden Instruments, London, UK). The separated slices were transferred into a prerecording chamber containing modified aCSF at room temperature and continuously bubbled with Carbox. Slices were maintained under these conditions at least 1 h prior to use, allowing for good tissue recovery.

## 2.6. Nitric Oxide and Oxygen Recording

Individual slices were placed in a recording chamber (BSC-BU with BSC-ZT top; Harvard Apparatus) and perfused with normal aCSF at 32 °C (temperature controller model TC-202A; Harvard Apparatus) bubbled with Carbox. The electrodes were inserted into the pyramidal cell layer of the CA1 subregion of the rat hippocampal slice, a region that is easy to identify and with a high expression of glutamatergic receptors and nNOS. The hippocampal slices were stimulated with NMDA (100 μM) or glutamate (5 mM) added to the perfusion aCSF for 2 min.

For OGD studies, slices placed in the recording chamber were perfused with aCSF without glucose and saturated with a gas mixture of 95% of nitrogen gas ($N_2$) and 5% $CO_2$ for 10 min. Then, reoxygenation was performed by replacing the perfused medium with normal aCSF saturated with Carbox.

To measure the pH variation during OGD, a pH microelectrode ($\varnothing$ = 100 μm, Unisense, Denmark) and an $O_2$ microelectrode were inserted in an antiparallel orientation into the pyramidal cell layer of the CA1 subregion of the rat hippocampal slice.

## 2.7. Data Analysis

Using the OriginPro 7.5 software (OriginLab Corporation, Northampton, MA, USA), the recorded $^\bullet$NO signals were individually analyzed in terms of (1) the $^\bullet$NO peak amplitude of the signal; (2) the signal area, calculated as the time integral of the $^\bullet$NO signal; and (3) the $^\bullet$NO signal width as the time in seconds from the stimulation point to return to basal levels.

All statistical analyses were performed using GraphPad Prism 5 software (GraphPad Software, San Diego, CA, USA). Data are presented as mean ± SEM. Statistical analysis of the data was performed using one-way analysis of variance (ANOVA) followed by the post hoc Bonferroni's multiple comparison test. Differences were considered significant at $p < 0.05$.

## 3. Results

### 3.1. Nitric Oxide Concentration Dynamics in Rat Hippocampus Slices Evoked by NMDA Stimulation of Neuronal Activity: The Modulatory Role of Nitrite

The hippocampal slices were stimulated with NMDA, the synthetic and specific agonist of the NMDA receptor (NMDAr), present in the perfusion medium for a 2 min period in the absence (control) and presence of three different concentrations of nitrite (100 μM, 500 μM and 2 mM). Ascorbate (500 μM) was always present in the aCSF during all the perfusions. This concentration of ascorbate was selected on the basis of our previous studies showing the functional coupling of ascorbate and nitric oxide dynamics in the rat hippocampus upon stimulation of NMDAr that induces the release of ascorbate from stimulated neurons, achieving a local extracellular concentration of about 500 μM [33,37]. Stimulation of the NMDAr causes a local oxygen tension drop (data not shown) and induces a biphasic $^\bullet$NO production in the CA1 subregion (Figure 1A). As observed in Figure 1A, the first component of the signal temporally coincides with the period of neuronal stimulation. Notably, in the presence of nitrite and in a concentration-dependent manner, both components of the $^\bullet$NO signal exhibit a greater amplitude (Figure 1A,B) and a greater area (Figure 1A,C). Regarding the $^\bullet$NO signal width, no significant differences are observed between the control condition and the presence of nitrite (Figure 1A,D).

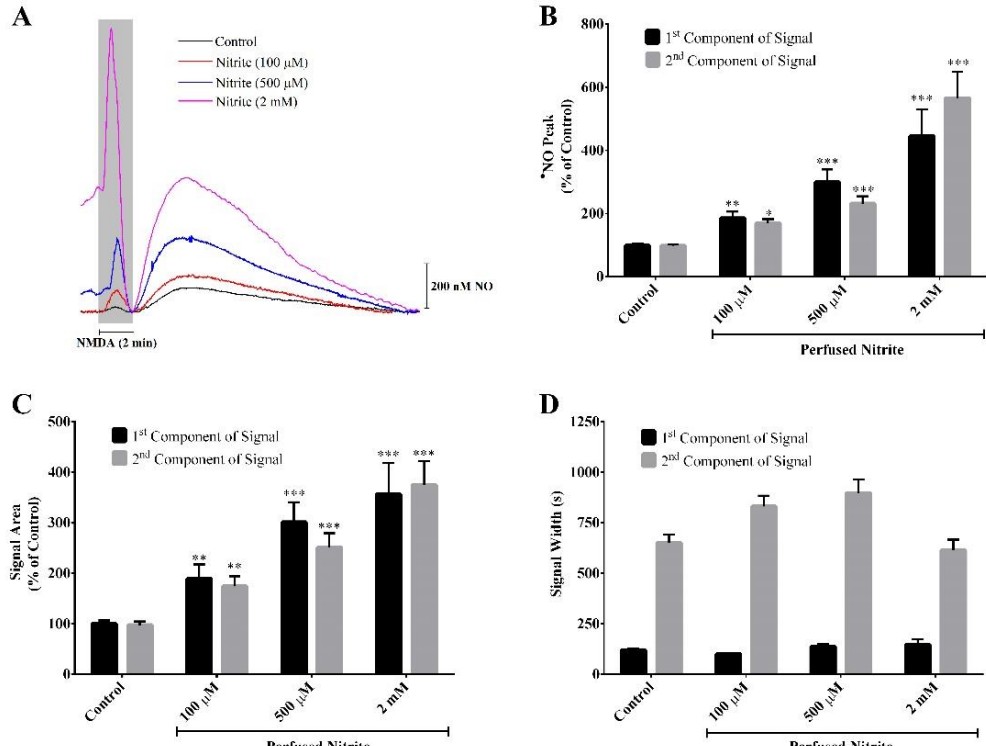

**Figure 1.** Nitrite enhances the production of •NO evoked by perfusion of NMDA (100 μM) for 2 min (shade area) in the presence of ascorbate (500 μM). Recordings were performed in the CA1 subregion of rat hippocampal slices. Ascorbate is present in the perfusion medium. (**A**) Representative amperometric recording of the •NO production in the absence (control) and presence of different nitrite concentrations (100 μM, 500 μM and 2 mM). (**B**) Amplitude, (**C**) area and (**D**) duration of the biphasic •NO signal. The first and the second components of the signal correspond to the first and the second peak of the biphasic •NO signal, respectively. Each bar represents the mean ± SEM. Statistical significance: * $p < 0.05$, ** $p < 0.01$ and *** $p < 0.001$ as compared with the control.

### 3.2. Nitric Oxide Concentration Dynamics in Rat Hippocampus Slices upon Stimulation of Neuronal Activity by Glutamate

Hippocampal slices perfused with aCSF containing ascorbate 500 μM were also stimulated with glutamate, the endogenous agonist of glutamate receptors, in the absence (control) and presence of nitrite (100 μM). It was found that, in the presence of nitrite, the signals corresponding to the production of •NO have a greater amplitude (Figure 2A,B), a greater area (Figure 2A,C) and a longer duration (Figure 2A,D). The •NO concentration dynamics are distinct from that obtained with the artificial NMDA receptor agonist (NMDA), notably the absence of a biphasic signal, likely reflecting more complex mechanisms of activation that translate into different kinetics of •NO production by the neuronal •NO isoform associated with the NMDA receptor.

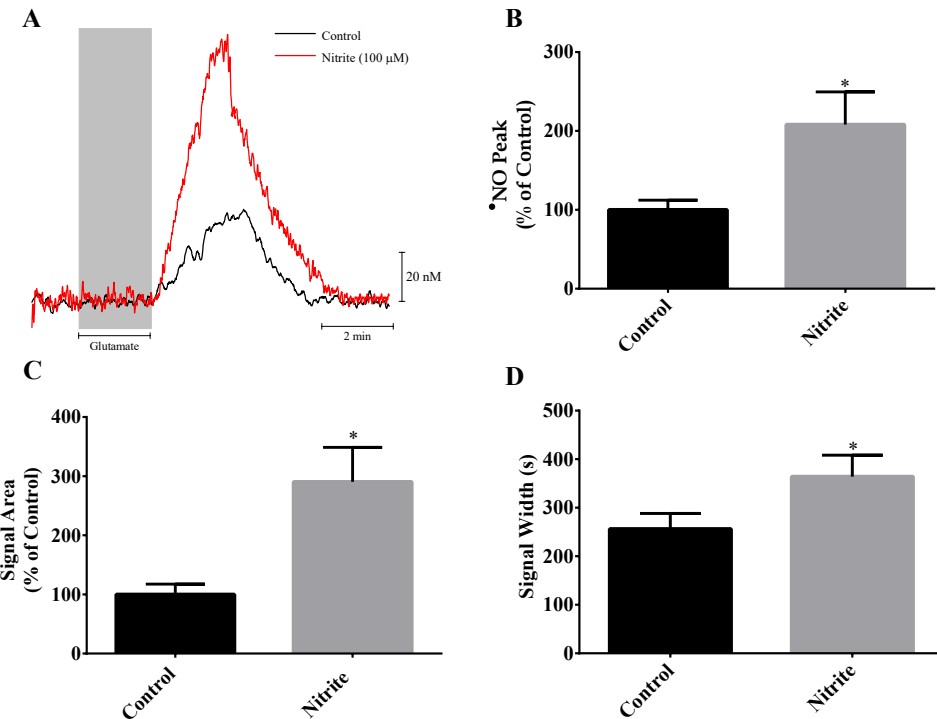

**Figure 2.** Nitrite enhances the production of $\bullet$NO evoked by perfusion of glutamate (5 mM) for 2 min in the presence of ascorbate (500 μM). Recordings were performed in the CA1 subregion of rat hippocampal slices. Ascorbate is present in the perfusion medium. (**A**) Representative amperometric recording of the $\bullet$NO production in the absence (control) and presence of nitrite (100 μM). (**B**) Amplitude, (**C**) area and (**D**) duration of the $\bullet$NO concentration dynamics. Each bar represents the mean $\pm$ SEM. Statistical significance: * $p < 0.05$ as compared with the control.

### 3.3. Evaluation of pH and Nitric Oxide Production in Rat Hippocampus Slices in Transient Oxygen-Glucose Deprivation (OGD)

The production of $\bullet$NO via the reduction of nitrite may be particularly relevant under hypoxic and acidotic conditions, a situation in which the enzymatic $\bullet$NO synthesis might be impaired as $\bullet$NO synthases use $O_2$ as a substrate. Therefore, using both a pH microelectrode and a selective CFM for $O_2$ measurement, pH and $O_2$ variations were simultaneously monitored in rat hippocampal slices subjected to hypoxia for 10 min, followed by reoxygenation. It is observed that, upon OGD conditions, the pH value gradually decreases from 7.4 during the hypoxia period, reaching a minimum of approximately 7.17 (Figure 3A). About 1.5 min after the start of reoxygenation, the pH gradually increases until it reaches the normal value of 7.4 (Figure 3A).

Also, $\bullet$NO and $O_2$ were simultaneously measured under OGD conditions in the absence (control) and presence of nitrite (100 μM, 500 μM and 2 mM) followed by reoxygenation. In the absence of added nitrite, we observe a signal detected by the $\bullet$NO microsensor during OGD and a further residual signal following NMDA (100 μM) stimulation for 2 min (Figure 3B). Both signals increase as a function of the nitrite concentration in the perfusion media, from 100 μM (Figure 3C), 500 μM (Figure 3D) to 2 mM (Figure 3E). In order to verify if this signal corresponded to $\bullet$NO production, the experiments were repeated under an applied potential of 0.4 V to the CFM, a potential at which oxidation of $\bullet$NO does not occur. Under these conditions, the signal almost disappears (Figure 3F). Figure 4 shows the quantification of the variations shown in Figure 3. It is of note that the signals obtained upon stimulation with NMDA (Figure 3B–E) under OGD conditions are smaller as compared with $\bullet$NO production observed in hippocampal slices not subjected to OGD (Figure 1).

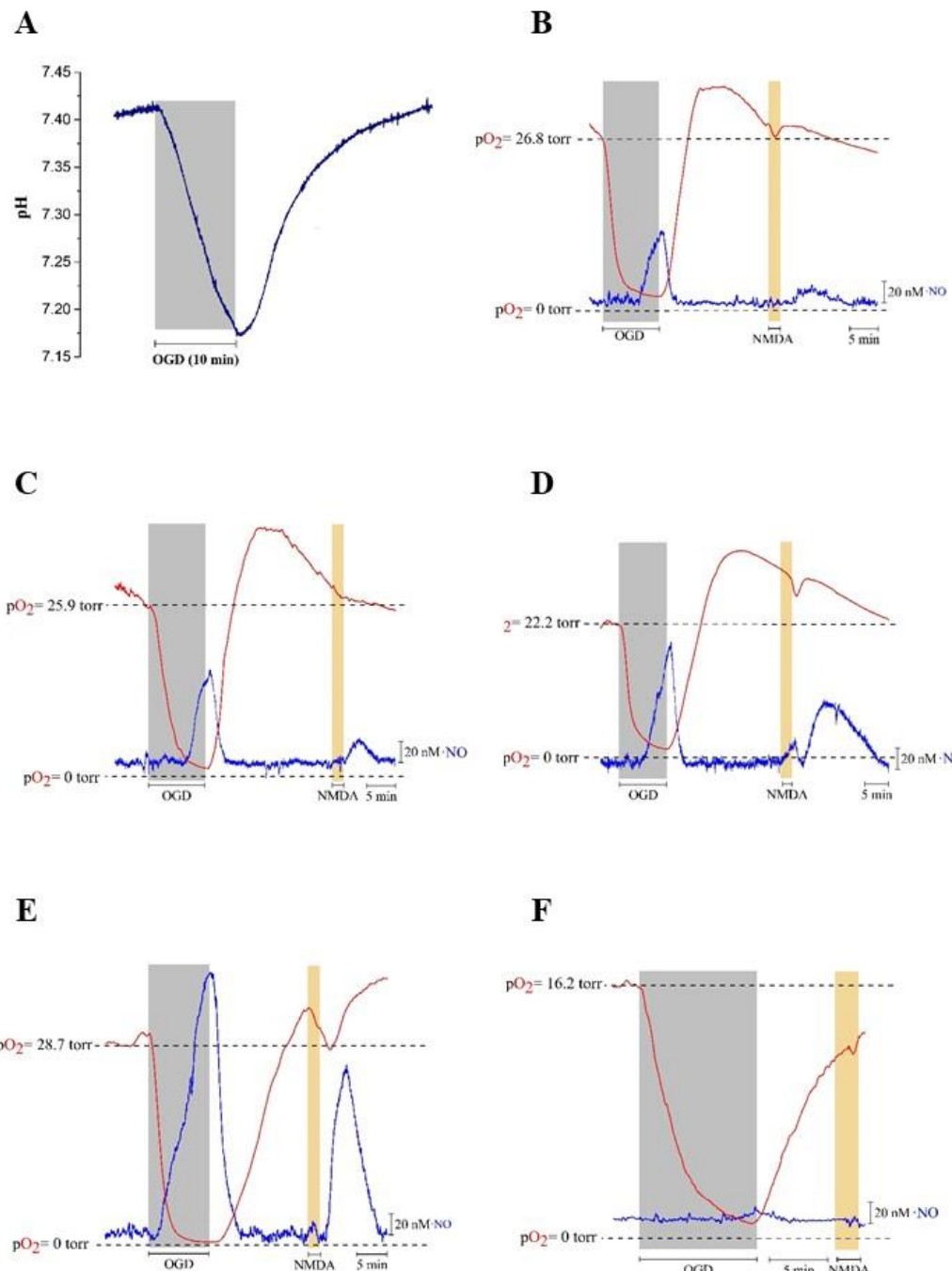

**Figure 3.** Nitrite enhances •NO production during OGD for 10 min and by perfusion of NMDA (100 μM) for 2 min after reoxygenation. Recordings were performed in the CA1 subregion of rat hippocampal slices. Ascorbate (500 μM) is present in the perfusion medium. (**A**) Representative recording of pH inside of the rat hippocampal slice during OGD for 10 min. Simultaneous recording of •NO (blue line) and $O_2$ (red line) in the (**B**) absence of nitrite (control), presence of (**C**) 100 μM nitrite, (**D**) 500 μM nitrite, (**E**) 2 mM nitrite and (**F**) absence of nitrite at 0.4 V.

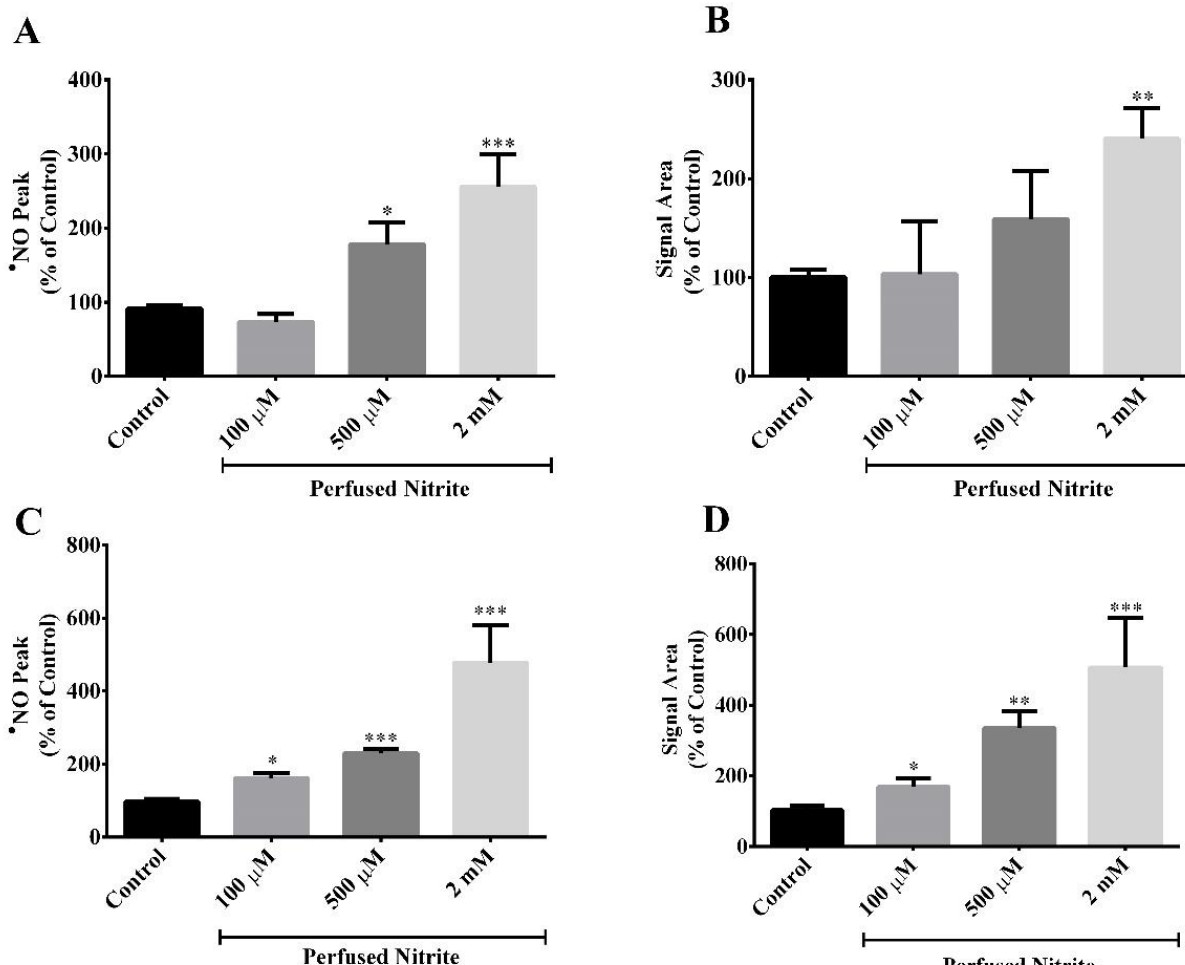

**Figure 4.** Quantification of $^\bullet$NO production during OGD in the presence of nitrite and by perfusion of NMDA (100 µM) for 2 min after reoxygenation. Recordings were performed in the CA1 subregion of rat hippocampal slices. Ascorbate (500 µM) is present in the perfusion medium. (**A**) Amplitude and (**B**) area of the $^\bullet$NO signal evoked by OGD. (**C**) Amplitude and (**D**) area of the $^\bullet$NO signal evoked by perfusion of NMDA (100 µM) for 2 min after reoxygenation. Each bar represents the mean $\pm$ SEM. Statistical significance: * $p < 0.05$, ** $p < 0.01$ and *** $p < 0.001$ as compared with the control.

## 4. Discussion

While the synthesis of $^\bullet$NO by NOS is $O_2$ dependent, the nitrate–nitrite–$^\bullet$NO pathway is gradually activated as tensions of $O_2$ drop, seeming to play a key role in vasodilation [38], modulation of mitochondrial respiration [39], hypoxia [40] and also in the tissue protection in ischemia-reperfusion [41].

Although the one-electron reduction of nitrite to $^\bullet$NO by dietary reductants is well established in the gastric compartment [22,23], this pathway of $^\bullet$NO production has been poorly explored in the brain environment. In this context, it is worth mentioning that ascorbate is a powerful reducing agent that exists in abundance in the brain, particularly in the cerebral cortex and hippocampus, where it may reach concentrations around 10 mM in neurons [42–44]. Moreover, there is evidence for functional coupling of ascorbate and $^\bullet$NO dynamics in the rat hippocampus upon stimulation of NMDAr [33], triggering ascorbate release from neuronal cells to the extracellular fluid during glutamatergic activity, probably by the glutamate–ascorbate heteroexchange mechanism [45–47], with ascorbate concentrations of up to 500 µM reported in the cerebrospinal fluid [42–44] and the extracellular neuronal environment [37].

Under hypoxic or ischemic conditions, such as in stroke, ischemia and aging, the NOS activity is impaired, the local pH drops and the reduction of nitrite to $^\bullet$NO may

constitute an important rescuing or alternative mechanism to the classical NOS pathway under these conditions [19,40]. The hippocampus seems to be the cerebral area most susceptible to hypoxic injury [48,49]. Importantly, several years ago, Millar proposed the reduction of nitrite to $^{\bullet}$NO in the extracellular space by ascorbate released from neurons under increased activity as a regulatory mechanism of vascular oxygen delivery according to the local metabolic needs of neurons [20]. Indeed, as neuronal stimulation causes local $O_2$ drop to very low tensions [50,51] (creating a "hypoxia-like" transient status), the reduction of nitrite to $^{\bullet}$NO by extracellular ascorbate concomitantly released from neurons is known to be favored under these conditions [52]. Although some studies showed that, under hypoxia, the pH might drop to values near 6.4, causing severe brain acidosis [53–55]. Here, we observed that OGD for 10 min induced a mild brain acidosis in the CA1 region of rat hippocampal slices, as the pH fell from 7.4 to 7.17. This may be justified by differences regarding the model of this study, the age of the animals and the OGD time.

In agreement with Millar's hypothesis, here we show that during conditions simulating ischemia-reperfusion conditions (transient OGD) and in the absence of neuronal nitric oxide synthase (nNOS) activation, a transient increase of $^{\bullet}$NO is observed in a nitrite concentration-dependent manner and in ascorbate-containing perfusion media (Figure 3). Residual nitrite contained in the slices might account for the $^{\bullet}$NO signal under control conditions (Figure 3B). In agreement with these results, it has been reported that there is a protective action of nitrite during ischemia and ischemia-reperfusion in several organs, including in the brain [56–58], and this protection is attributed to its reduction in $^{\bullet}$NO.

These observations strongly support nitrite as the source of $^{\bullet}$NO under conditions in which nNOS might be inoperative due to diminished $O_2$ bioavailability. Moreover, it is of note that, following OGD and under normal conditions, the stimulation with NMDA induces $^{\bullet}$NO transients that, although expected, are quantitatively smaller as compared with the OGD for each of the recordings (Figure 3B–E).

Expectedly, different kinetic traces are obtained with NMDA and glutamate that likely reflect the complexity of the pathways leading to NMDAr activation by glutamate as compared with NMDA, but, as in the case of NMDA stimulation, the glutamate-dependent $^{\bullet}$NO transients are dependent on the concentration of nitrite in the perfusion medium.

It should be mentioned that controls in the absence of ascorbate in the perfusion media in all the experiments induced little quantitative effects of NO concentration dynamics. As mentioned before, our previous studies supported a functional coupling of ascorbate and $^{\bullet}$NO dynamics in the rat hippocampus upon stimulation of NMDAr that induces the release of ascorbate from stimulated neurons [33,37]. In fact, considering the millimolar range of ascorbate in neurons, the concentration of ascorbate released into the extracellular medium upon glutamatergic stimulation could be extremely high and, thereby, further increasing ascorbate concentration via its addition to the perfusion medium did not translate into an increase of nitrite reduction. Furthermore, some specific enzymes, such as hemoglobin [39,59], xanthine oxidoreductase [60,61], complexes of the mitochondrial electron transport chain [62–64] and even the NOS [65], seem to acquire a nitrite reductase activity, particularly at low $O_2$ tension, and may be involved in the $^{\bullet}$NO production from nitrite. Thus, we wanted to make sure the presence of enough ascorbate to reduce nitrite in all the several conditions, including those of OGD where ascorbate release from neurons is not coupled to NMDAr activation.

The concentrations of nitrite used in this study (100 μM, 500 μM and 2 mM) might be considered exceedingly high concentrations. However, in our model of study, hippocampal slices were perfused with the aCSF that contains nitrite, with a nitrite gradient along the thickness of the slice being expected. Because the recordings are performed in the core of the slice, we chose these concentrations to ensure that an adequate concentration of nitrite reached the core of the slice, local to where the microsensors were inserted. Moreover, these concentrations permit to clearly evaluate whether the effect of nitrite was concentration-dependent.

## 5. Conclusions

Overall, this work strongly suggests that nitrite is a precursor of •NO in the brain in a way that is functionally dependent on glutamate NMDAr activation via the NMDAr–nNOS pathway but also under conditions in which nNOS might be impaired, notably hypoxia. Given the critical role of •NO as the direct mediator of neurovascular coupling (NVC [7], a process critical for brain structure and function [66]), the mechanism of •NO production from nitrite via its reduction to •NO by ascorbate may represent a key physiological mechanism that participates in the regulation of brain homeostasis by improving the NVC and CBF and could acquire a particular relevance in situations where NVC is compromised, such as ischemia, neurodegeneration and aging. Therefore, the nitrite–•NO pathway may open avenues to the development of new and innovative therapies to improve the impaired NVC observed in several conditions, including neurodegenerative diseases, aging and diabetes [55,66,67].

**Author Contributions:** C.N.: conceptualization, formal analysis, investigation, and writing original draft preparation. J.L.: conceptualization, supervision, funding acquisition, and writing—review. All authors have read and agreed to the published version of the manuscript.

**Funding:** This work was financed by the European Regional Development Fund (ERDF) through the COMPETE 2020 (Operational Programme for Competitiveness and Internationalization) and by the Portuguese National Funds via FCT—Fundação para a Ciência e Tecnologia—under projects POCI-01-0145-FEDER-029099, 2022.05454.PTDC, UIDB/04539/2020, UIDP/04539/2020 and LA/P/0058/2020.

**Institutional Review Board Statement:** The animal study protocol was approved by the local ethics committee (ORBEA 146_2016/31102016) and the Portuguese Directorate-General for Food and Veterinary.

**Informed Consent Statement:** Not applicable.

**Data Availability Statement:** The data presented in this study are available upon request from the corresponding author.

**Acknowledgments:** Authors are thankful to Cátia F. Lourenço for their help in the preparation of the microelectrodes.

**Conflicts of Interest:** The authors declare no conflict of interest.

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
