# Peer review of "Nitric Oxide Production from Nitrite plus Ascorbate during Ischemia upon Hippocampal Glutamate NMDA Receptor Stimulation"

_2673-6411, doi:10.3390/biochem3020006_

Round 1

Reviewer 1 Report

The manuscript submitted by Nunes et al reports the generation of nitric oxide in rat hippocampal slices from nitrite via reduction in the presence of ascorbate. This manuscript is well structured overall, but it needs minor modifications, that I have outlined in the comments below.

1.     Similar work was already published by the authors, however in this manuscript they have performed the reduction of nitrite to nitric oxide in the presence of ascorbate.

2.     I appreciate the authors work for evaluating the pH in transient oxygen and glucose deprivation.

3.     Figure 1: What is 1st and 2nd component of signal? Please clarify. In Fig. 1A, scale on X-axis is confusing. 

4.     Line 208: What does the authors mean by reflecting different kinetics? Please elaborate.

5.     Line 337: The concentration of nitrite? Do they mean Sodium nitrite?

6.     Conclusion is missing. Please add this section.

7.     This paper would have greater impact if the real-time measurements of nitrite were done in the live brain extracellular space using microelectrode sensor?

8.     If possible please change the manuscript title.

9.     The authors have to proofread the manuscript, there are grammatical errors.

Reviewer 2 Report

Manuscript Number: biochem-2235242

Title: Nitric oxide is produced in rat hippocampal slices from nitrite in the presence of ascorbate via NMDA receptor activation and under-model ischemic conditions

Journal: Journal of BioChem

The authors have investigated the NO production in concentration-dependent manners and in the presence of ascorbate evoked by NMDA and glutamate stimulation of rat hippocampal slices. Also, nitrite potentiates the NO production induced by oxygen and glucose deprivation. The authors have provided a promising prospect in the potential clinical impact of the notion of a redox interaction of ascorbate with nitrite yielding NO upon neuronal glutamatergic activation and given the critical role of NO as the direct mediator of neurovascular coupling may represent a key physiological mechanism by which NO production for cerebral blood flow responses to neuronal activation is sustained under hypoxic/acidic conditions in the brain. However, some details need further explanation. The article is well written and well organized, and I believe that it is suitable for publication in the Journal of BioChem. However, I recommend this manuscript can be published after a minor revision and improvement.

1. Title

The title should be descriptive instead of wordy, such as "Nitric oxide is produced in rat hippocampal slices from nitrite in the presence of ascorbate via NMDA receptor activation and under-model ischemic conditions". In this regard, such terms should be changed with more appropriate terms or deleted to be more specific to a point. The terms, such as "produced or production", "in the presence of," and "under-model ischemic conditions or ischemic model or so on with specific terms".

2. Abstract

1. The authors need to be consistent regarding the abbreviate terms that should be expressed in their first expression throughout the manuscript. Here, the abbreviate and form of such terms have been used more than on the first expression, such as Nitric oxide (•NO) and deprivation of oxygen and glucose (OGD).

2. The abbreviation should be "oxygen-glucose deprivation (OGD)" and should be consistent throughout the manuscript.

3. The author should confirm what the black dot represents with the "Nitric oxide (•NO)". However, we studied, but such a new method of writing is out of our knowledge.

3. Introduction

1. The statement should be; "In this work, using microsensors with a high degree of selectivity to NO, we explore the non-NOS production of NO via nitrite reduction in rat hippocampal slices stimulated with NMDA and glutamate and also subjected to oxygen and glucose deprivation (OGD)".

4. Materials and Methods

1. 2.1. Materials and Reagents

1. The authors need to be consistent regarding the abbreviate terms that should be expressed in their first expression throughout the manuscript. Here, the abbreviate and form of such terms have been used more than on the first expression, such as N-Methyl-D-aspartate (NMDA).

2. The statement should be "In order or so on to improve the viability of the slices." However, writing "To" at the start of the sentence makes grammar errors in such phrases.

5. Results

1. 3.1 Nitric oxide concentration dynamics in rat hippocampus slices evoked by NMDA stimulation of neuronal activity: the modulatory role of nitrite

1. In Figure 1, part D, like the A-C, should be cited to make consistency.

2. The author should confirm heading number "3. 3" has not been arranged with a sequence.

6. Conclusion

The author should write a separate paragraph with "5. Conclusion/Summary" at the end of the manuscript to make it easier for future readers.

7. Please correct typos, errors, and grammar throughout the manuscript; also, the manuscript needs consistency regarding writing and formatting.
